# Research Progress on the Mechanism of Milk Fat Synthesis in Cows and the Effect of Conjugated Linoleic Acid on Milk Fat Metabolism and Its Underlying Mechanism: A Review

**DOI:** 10.3390/ani14020204

**Published:** 2024-01-08

**Authors:** Yuanyin Guo, Ziang Wei, Yi Zhang, Jie Cao

**Affiliations:** 1College of Veterinary Medicine, China Agricultural University, Beijing 100193, China; guoyuanyin979@163.com (Y.G.); m13581872198@163.com (Z.W.); 2College of Animal Science and Technology, China Agricultural University, Beijing 100193, China; yizhang@cau.edu.cn

**Keywords:** CLA, dairy cow, milk fat, mechanism

## Abstract

**Simple Summary:**

Although fat does not constitute the largest proportion of cow milk composition, the energy consumed in milk fat production is the highest among all milk components. Milk fat can improve the flavor of milk, and it is an indicator of the nutritional value of milk; therefore, exploring the synthesis of milk fat in dairy cows has become a hot topic. The classic mechanism of milk fat biosynthesis involves the de novo synthesis of fatty acids (fatty acid synthase, acetyl-CoA carboxylase, etc.—abbreviated as FASN, ACACA, etc.); the uptake, transport, and activation of long-chain fatty acids (lipoprotein lipase, long-chain acyl-CoA synthetase, etc.—abbreviated as LPL, ACSL, etc.); the synthesis of triglycerides (1-acylglycerol-3-phosphate O-acyltransferase 6, Diacylglycerol acyltransferase, etc.—abbreviated as AGPAT6, DGAT, etc.); signaling pathways; and transcription factors (mechanistic target of rapamycin, sterol regulatory element-binding protein, peroxisome proliferator-activated receptor gamma, and adenosine monophosphate-activated protein kinase—abbreviated as mTOR, SREBP, PPARG, and AMPK). The inhibitory effect of conjugated linoleic acid (CLA) on milk fat synthesis in cows has been demonstrated in many studies, including in vitro and in vivo experiments. However, the impact of CLA on milk fat is limited to a single gene or pathway. This article reviews the possible relationships between various genes, pathways, and transcription factors. New synthesis mechanisms, such as thyroid hormone-inducible hepatic protein, methyltransferase complex containing methyltransferase-like 3, elongation of very long-chain fatty acid protein, and phosphatidate phosphatase 1 (abbreviated as THRSP, METTL3, ELOVL, and LPIN1), and new candidate genes involved in milk fat biosynthesis, including phosphatidylinositol 4-kinase type 2 alpha, solute carrier family 16 member 1, ATPase phospholipid transporting 8A2, vascular endothelial growth factor D, an inhibitor of DNA binding 1, and arachidonate 12-lipoxygenase (abbreviated as PI4K2A, SLC16A1, ATP8A2, VEGFD, ID1, and ALXO12), provide references for potential mechanisms of CLA-mediated regulation of milk fat metabolism.

**Abstract:**

Milk fat synthesis in cows mainly includes the synthesis of short- and medium-chain fatty acids, the uptake, transport, and activation of long-chain fatty acids (LCFAs), the synthesis of triglycerides, and the synthesis of the genes, transcription factors, and signaling pathways involved. Although the various stages of milk fat synthesis have been outlined in previous research, only partial processes have been revealed. CLA consists of an aggregation of positional and geometric isomers of linoleic fatty acid, and the accumulated evidence suggests that the two isomers of the active forms of CLA (cis-9, trans-11 conjugated linoleic acid and trans-10, cis-12 conjugated linoleic acid, abbreviated as c9, t11-CLA and t10, c12-CLA) can reduce the fat content in milk by regulating lipogenesis, fatty acid (FA) uptake, oxidation, and fat synthesis. However, the mechanism through which CLA inhibits milk fat synthesis is unique, with most studies focusing only on the effects of CLA on one of the genes, transcription factors, or signaling pathways involved. In this study, we summarized the structure and function of classic genes and pathways (mTOR, SREBP, AMPK, and PPARG) and new genes or pathways (THRSP, METTL3, ELOVL, and LPIN1) involved in each stage of milk fat synthesis and demonstrated the interactions between genes and pathways. We also examined the effects of other substances (melanin, nicotinic acid, SA, etc.). Furthermore, we evaluated the influence of β-sitosterol, sodium butyrate, Met arginine, and *Camellia oleifera* Abel on milk fat synthesis to improve the mechanism of milk fat synthesis in cows and provide a mechanistic reference for the use of CLA in inhibiting milk fat biosynthesis.

## 1. Introduction

Dairy cows are economically important animals, and they constitute a vital component of the world’s animal husbandry industry. Milk is rich in protein, fat, trace elements, and vitamins essential for human physiology. The protein and fat contents of cow milk are crucial factors in evaluating milk quality [1].

Milk fat plays a key role in the efficiency of dairy products and pasture production. The mechanism of milk fat biosynthesis has become increasingly clear. CLA serves an inhibitory role in milk fat synthesis. Specifically, t10, c12-CLA is a biologically active FA that can reduce milk fat synthesis by downregulating genes involved in the synthesis of breast fat [2,3,4]. CLA consists of an aggregation of geometric isomers with 18 carbon atoms and conjugated double bonds in their configuration. As such, CLA is categorized as a long-chain polyunsaturated fatty acid. The double bonds in CLA are usually in a cis configuration, and they are located on the 9th and 12th carbon atoms. CLA has several physiological effects, including anticancer, immunomodulatory, and antioxidant effects [5,6], in addition to decreasing cholesterol levels, reducing the incidence of cardiovascular diseases [7,8], and inhibiting fat deposition [9,10]. The two main approaches used for milk fat production are endogenous synthesis and synthetic production. In endogenous synthesis, stearoyl-CoA desaturase (SCD) acts upon the hydrogenation intermediate trans-11 C18:1 of unsaturated fatty acids in the rumen to synthesize CLA. In synthetic production, both chemical and biological methods may be used for milk fat production. The use of different methods and raw materials leads to differences in the purity of the produced CLA. To date, several isomers of CLA, such as t9, t11, c9, c11, t10, t11, c10, and c12 CLA, have been identified, among which c9, t11, t10, and c12 CLAs are the most bioactive [11]. In this study, we attempt to provide a brief overview of the existing network of mechanisms involved in milk fat synthesis, providing a reference for the mechanism through which CLA regulates milk fat biosynthesis in bovine mammary epithelial cells (BMECs).

## 2. Anabolism of Milk Fat

Milk fat biosynthesis in ruminants has been extensively studied, and it is primarily achieved through the synthesis of triglycerides (TGs) from glycerol and fatty acids. Mechanistically, the sn-1 position of glycerol 3-phosphate is catalyzed by glycerol-3-phosphate acyltransferase, mitochondrial (GPAM), to generate lysophosphatidic acid (LPA). Subsequently, 1-acylglycerol-3-phosphate O-acyltransferase 6 (AGPAT6) catalyzes the sn-2 position of LPA to produce phosphatidic acid (PA). Phosphatidate phosphatase LPIN (LIPIN) then cleaves the phosphate group from PA to form diacylglycerol (DAG), which is eventually catalyzed by diacylglycerol acyltransferase (DGAT) to generate TGs [12,13]. All of the aforementioned fatty acids can be classified as short-, medium-, or long-chain fatty acids, each stemming from different sources. While short- and medium-chain FAs are primarily produced through de novo synthesis in mammary epithelial cells, LCFAs are mostly derived from non-esterified fatty acids present in the plasma. (1) The first step in the de novo synthesis of FAs in ruminants involves the fermentation of carbohydrates in the rumen. During fermentation, acetate and butyrate are produced, and the latter is further oxidized to generate β-hydroxybutyrate. FAs are then biosynthesized de novo from the acetate and β-hydroxybutyrate produced in the previous steps by acetyl-CoA carboxylase (ACACA) and fatty acid synthase (FASN). The primary process of de novo synthesis involves the ACACA-mediated catalysis of acetyl-CoA to produce malonyl-CoA, followed by the catalytic action of fatty acid synthase on acetyl-CoA and malonyl-CoA, which ultimately generates fatty acids [14]. (2) The uptake, transport, and activation of FAs involve the following steps: Mammary cells are responsible for the uptake of FAs from the blood, after which these fatty acids are hydrolyzed by the action of the very low-density lipoprotein receptor (VLDLR) and lipoprotein lipase (LPL). The hydrolyzed FAs then enter mammary epithelial cells through free diffusion or active transport. Inside the cells, fatty acids undergo activation via attachment to long-chain acyl-CoA synthetase (ACSL), which is bound to fatty acid binding protein 3 (FABP3) [15,16] and are then conveyed to the site where they are used as substrates of stearoyl-CoA desaturase (delta-9-desaturase, SCD) for the synthesis of TG.

### 2.1. Mechanistic Target of Rapamycin (mTOR)

Previous studies have demonstrated that mTOR regulates both the milk protein synthesis [17] and the lipid metabolism [18]. According to the work of Zhao et al. [19] and Osorio et al. [20], mTOR may also serve as a key node in the biosynthesis of lactose, milk protein, and milk fat.

The interactions of mTOR usually involve the formation of the following two complexes: mechanistic targets of rapamycin complexes 1 and 2 (mTORC1 and mTORC2) [21]. The principal role of mTORC1 is to integrate growth factors, energy status, oxygen delivery, and amino acid utilization to regulate several processes that are involved in the promotion of cell metabolism. The primary action of mTORC2 is to serve as a target responsive to growth factors and a regulator of cell metabolism [21,22,23]. It is speculated that during translation, mTORC1 governs cell metabolism through the rapamycin-sensitive phosphorylation of two key downstream substrates, S6 kinase 1 (S6K1) and eIF4E-binding protein 1 (4E-BP1) [24,25,26,27]. The mechanical targets of mTORC2 primarily act as effectors of the insulin/phosphoinositide 3-kinase (PI3K) signaling pathway to control cell proliferative activity; furthermore, they are correlated with cellular metabolic regulation. In addition to regulating protein production, mTOR regulates lipid metabolism [28,29]. In cattle, milk synthesis is considered a nutrient-central regulator of growth factors, and it appears to be governed by a complex interaction between peroxisome proliferator-activated receptor gamma (PPARG) and other transcription factors [20]. Along with other genes or pathways related to fat synthesis, mTOR also governs the biosynthesis of milk fat in cattle.

The mTORC1 complex comprises mTOR, raptor, GβL, and deptor, while the mTORC2 complex comprises mTOR, RICTOR, GβL, PRR5, deptor, and SIN1. The mTORC1 complex (RAPTOR) is recognized as the principal conductor of cell metabolism, and it is responsible for accumulating triacylglycerols in sufficient quantities for the de novo biosynthesis of FAs. In contrast, the role of mTORC2 (RICTOR) in lipid metabolism is yet to be outlined. In separate studies, Anderson et al. and Luyimbazi et al. blocked the expression of the target genes of sterol regulatory element-binding protein 1 (SREBP1), i.e., FASN, ACACA, and stearoyl-CoA desaturase-1 (SCD1), using rapamycin, a specific inhibitor of mTORC1, thereby demonstrating the involvement of mTORC1 in FA biosynthesis [30,31]. Liu et al. [32] adopted the induction strategy of using acetate salts and reported that AGPAT6 could produce PA to activate mTORC1 signal transduction during milk fat synthesis. As a transcription factor, PPARG can mediate the effect of acetate on AGPAT6 via Retinoid X receptor α (RXRα), which indicates that mTORC1 is regulated at the upstream region through PPARG and AGPAT6 for milk fat synthesis. Previous studies have demonstrated that mTORC1 positively modulates SREBP1 activity [33,34]. However, rapamycin does not affect the expression of any of the target genes of SREBP in all cellular contexts [35], which indicates that mTORC1 promotes SREBP function. SREBP is a prime modulator of lipid and sterol gene transcription. The mechanism through which mTOR regulates SREBP was elucidated by Peterson TR et al. Treatment with an mTOR inhibitor resulted in altered cellular localization and nuclear eccentricity of LPIN1, as well as the downregulation of SREBP protein levels [36]. These findings indicate that mTORC1 promotes milk fat biosynthesis by modulating the orientation of lipin 1 to modulate the SREBP pathway. According to previous studies, mTORC1 regulates the expression of fatty acid-synthesizing enzymes, such as ACACA, FASN, and SCD1, via the transcription factor SREBP [37,38]. The expression of both SREBP1 and SREBP2 was decreased in raptor-deficient T cells [39]. In hepatocytes, SREBP1c mRNA expression is increased through the mediating effect of insulin, and this increase is associated with mTORC1 but not S6K [34,40,41]. In summary, mTORC1 regulates SREBP at the transcriptional and protein expression levels to control adipogenesis. In adipose tissue, triacylglycerol lipase (ATGL) is among the crucial enzymes that catalyze triacylglycerol hydrolysis. Fat degradation is inhibited by mTORC1 through the inhibition of ATGL [42]. Studies have demonstrated that mTORC1 reduces the activity of ATGL by stimulating the expression of early growth response factor 1 (Egr1), a transcriptional inhibitor of ATGL [43]. The inhibition of mTORC1 increases fatty acid oxidation (FAO) and carnitine palmitoyltransferase I (CPT1) activity. CPT1 is a regulatory enzyme in the fatty acid oxidation process [44]. According to these studies, mTORC1 positively regulates fat production and inhibits fat degradation. The most common methylation of mRNA is m6A methylation, which is dynamically reversible and catalyzed mainly via a methyltransferase complex containing methyltransferase-like 3 and 14 (METTL3 and METTL14) [45] and Wilms’ tumor 1-associating protein (WTAP) [46]. METTL3 regulates adipogenesis by promoting the m6A methylation of mRNAs, such as Cyclin A2 (CCNA2), FASN, and Janus kinase 1 (JAK1) [47,48,49]. Wang et al. [50] studied the localization of METTL3 in BMECs and its effect on milk fat. The authors reported that METTL3 was located in the nucleus of BMECs and mediated the mRNA expression of mTOR and SREBP1 under activation by Met and E2, which are extracellular stimuli for lactating mammary epithelial cells. By controlling m6A methylation, METTL3 consequently modulates the mTOR and SREBP1 signaling pathways, ultimately facilitating the biosynthesis of milk proteins and fat.

Studies have revealed that the inhibition, knockout, and overexpression of mTORC2 exert important effects on lipid metabolism. In addition, a novel function of mTORC2 has been revealed: the regulation of brown adipocyte differentiation and a reduction in the expression of nuclear sterol regulatory element-binding protein 1c (nSREBP1c) and lipogenic genes in brown adipose tissue [51]. Li et al. [52] studied several cancer cell lines and reported that silencing RICTOR in these cells reduced the active form of SREBP1c, leading to decreased expression of the target genes ACC and FASN. In addition, inhibiting mTOR kinase or knocking out RICTOR in these cells induced the degradation of SREBP1c via the glycogen synthase kinase 3/F-box/WD repeat-containing protein 7 (GSK3/FBXW7) pathway, indicating that mTORC2 engages in regulating the stability of SREBP1c. Hagiwara et al. [53] and Yuan et al. [54] reported that specifically knocking out the RICTOR gene in mouse liver cells led to a loss of SREBP1c activity, which ultimately reduced lipogenesis. These mice retained the activity of functional mTORC1, indicating that mTORC2 specifically affects lipid synthesis. In addition, mTORC2 acts on protein kinase B (Akt). In separate studies, Yecies JL and Wan M et al. [40,55] demonstrated that Akt is necessary for adipogenesis. When Akt signaling was downregulated, mTORC1 activation alone was insufficient to stimulate adipogenesis. Yuan et al. [54] further confirmed this by demonstrating that even the upregulation of Akt signaling could not promote adipogenesis in mice with liver-specific RICTOR knockout. These studies demonstrated that mTORC2 and Akt are closely associated with adipogenesis. However, certain studies have elucidated the specific relationship through which mTORC2 phosphorylation catalyzes the phosphorylation of BSD domain-containing signal transducer and Akt interactor (BSTA), a protein containing the BSD domain, inducing BSTA to bind to Akt. This results in the generation of the BSTA–Akt complex, which subsequently inhibits Forkhead box C2 (FoxC2), a transcription inhibitor of the differentiation of white adipocytes [56]. Previous studies have demonstrated that mTORC2 controls insulin-stimulated glucose uptake and de novo lipogenesis (DNL) in adipocytes. However, when RICTOR was knocked out, most measured AKT substrates appeared to be phosphorylated normally [51,57]. These findings indicated that downstream AKT signaling remains unaffected upon mTORC2-RICTOR knockout. To explain this phenomenon, Martinez Calejman C et al. attempted to identify mTORC2-regulated AKT substrates and metabolites, including ATP citrate lyase (ACLY) and the acetyl-CoA generated by ACLY, using mass spectrometry. The authors reported that mTORC2 enhances the activity of carbohydrate response element-binding protein (ChREBP) and promotes the acetylation of histone proteins and glycolipid gene expression via ACLY. In addition, mTORC2 was shown to promote acetyl-CoA synthetase short-chain family member 2 (ACSS2)-mediated acetate synthesis and acetyl-CoA expression. Accordingly, it was inferred that AKT signaling mediated by mTORC2 in mTORC2-RICTOR knockout brown adipocytes is selective [58]. Studies have reported that mTORC2 enhances the de novo synthesis of FAs and triglyceride generation. Moreover, knocking out mTORC2-RICTOR in mammary epithelial cells inhibited the expression of the transcription factor PPARG and specific lipid-synthesizing genes. In addition, intracellular triglyceride accumulation and FA levels decreased [59]. Guo Z et al. reported that downregulating PPARG expression through the inhibition of RICTOR/mTORC2 reduced the expression of lipogenic genes, specifically LPIN1, diacylglycerol acyltransferase-1 (DGAT1), ACACA, and FASN [59]. These findings demonstrated that mTORC2 modulates the expression of lipid-related genes via PPARG, thereby promoting adipogenesis in vitro. In summary, mTORC2 exerts effects similar to those of mTORC1, involving the positive regulation of adipogenesis and inhibition of lipolysis; however, the two exert their effects via different pathways and mechanisms. The structure, functional description, and interaction with conjugated linoleic acid of mTOR are shown in Figure 1.

### 2.2. Sterol Regulatory Element-Binding Proteins (SREBPs)

SREBPs are a group of transcriptional regulators that modulate the transcription of major enzymes associated with the de novo biosynthesis of FAs, including ACLY, ACAA, FASN, and SCD enzymes. Three isoforms of SREBP proteins, namely sterol regulatory element-binding protein 1a, 1c, and 2 (SREBP1a, SREBP1c, and SREBP2), are expressed in mammalian cells and are encoded using the sterol regulatory element-binding transcription factor 1 and 2 (SREBF1 and SREBF2). SREBP1a and SREBP1c are transcribed from SREBF1 through the action of different promoters, while SREBP2 is encoded using the SREBF2 gene [60,61]. SREBP1 serves as a key transcription factor regulating lipogenesis [12]. It is mainly involved in the metabolism of FAs and TGs in vivo and the direct regulation of multiple lipid metabolism-related genes. However, SREBP2 regulates the cholesterol biosynthesis [61,62]. Li et al. [63] reported that SREBP1c overexpression markedly upregulated the activity of acyl-CoA synthetase long-chain family member 1 (ACSL1), leading to increased hepatic fatty acid uptake and activation. The authors also reported significantly increased triglyceride levels upon ACSL1 overexpression. In contrast, ACSL1 downregulation had opposite effects to those observed after overexpressing ACSL1 and the lipogenic pathway-associated genes SREBP1, ACC, FAS, and SCD1 [64]. In summary, ACC1, FAS, and SCD1 are target genes of SREBP1c, encoding key enzymes for lipid synthesis. In addition to associating with ACSL1, SREBP1 also acts on the acyl-CoA synthetase long-chain family member 4 (ACSL4), which belongs to the ACSL family. Chen et al. identified two regulatory relationships between SREBP1 and ACSL4 in hepatocellular carcinoma cells: (1) direct binding of c-Myc to the promoter region of the SREBP1 gene, resulting in transcriptional activation of SREBP1 expression; and (2) regulation of the stability of the SREBP1 protein through the F-box and WD repeat domain containing 7 (FBW7) [65]. FBW7 is the E3 ubiquitin ligase of SREBP1 [66]. The above study also demonstrated that ACSL4 could promote fatty acid synthesis in hepatocellular carcinoma cells by stimulating lipogenic enzymes, such as ACLY, ACC, FASN, and SCD. Previous studies have demonstrated that, in its inactive form, SREBP enters the SREBP cleavage-activating protein (SCAP)-SREBP protein complex with SCAP and is sequestered inside the endoplasmic reticulum through insulin-induced gene 1 (INSIG1) [67]. The INSIG protein is an anchor protein of the endoplasmic reticulum (ER) that has two isomers, namely INSIG1 and insulin-induced gene 2 (INSIG2) [68,69]. INSIG1 has a negative feedback effect on the SREBP signaling pathway [70]. SREBP activation occurs in two ways: (1) Phosphoenolpyruvate carboxykinase 1 (PCK1) binds to insulin-induced genes (INSIGs) to mediate INSIG phosphorylation, which subsequently inhibits the interaction between INSIGs and SCAP. The SCAP–SREBP protein complex is then separated from the endoplasmic reticulum and transported toward the Golgi apparatus. The Golgi membrane continuously releases two proteases, Site-1 and Site-2, to cleave SREBP and activate it. Active SREBP subsequently translocates into the nucleus and activates lipogenesis in an SREBP-dependent manner [71,72,73]. (2) Zeng et al. [74] reported that cluster of differentiation 36 (CD36), a fatty acid transporter, could bind directly to INSIG2, causing the SCAP/SREBP1 complex to separate from INSIG2 and be transported to the Golgi apparatus. SREBP is sequentially cleaved by Site-1 and Site-2 proteases, leading to lipogenesis. After being transported to the nucleus, SREBP specifically interacts with sterol regulatory elements (SREs) located in the promoter region of its target genes, thereby stimulating the genes associated with the biosynthesis of FAs, cholesterol, and triglycerides [75]. The structure, functional description, and interaction with conjugated linoleic acid of SREBPs are shown in Figure 2.

### 2.3. Adenosine Monophosphate-Activated Protein Kinase (AMPK)

Adenosine monophosphate-activated protein kinase (AMPK) is a serine/threonine protein kinase that is highly conserved in most eukaryotes. It comprises a catalytic subunit (α) and two regulatory subunits (β and γ) of several different types: α1, α2, β1, β2, γ1, γ2, and γ3 [76,77]. The AMPK molecule serves as a multifunctional enzyme that promotes the activity of different signaling pathways, including those associated with energy regulation. AMPK is considered a detector of the available energy in eukaryotic cells [78], and it is involved in apoptosis, proliferation, and autophagy [79,80]. Extensive research has revealed that AMPK plays different roles in different cells or tissues. In T cells, AMPK mainly regulates cell migration. In the hypothalamus, AMPK mostly regulates food intake [81,82]. AMPK activation in cells or tissues is regulated mainly through Ca^2+^ signaling and calmodulin-dependent protein kinase (CaMKK). Calmodulin-dependent protein kinase β (CaMKKβ) is the main factor responsible for phosphorylating Thr172, the α subunit site of AMPK [83], which is a prerequisite for the complete activation of AMPK [84]. The principal role of the β subunit of AMPK is to serve as a brace for the binding of the α and γ subunits to form a functional AMPK heterotrimer. The γ subunit is an adenine nucleotide binding site, and under low-energy conditions, ADP or AMP is produced in large quantities; these proteins bind to the γ subunit and alter the molecular architecture of AMPK, thereby promoting Thr172 phosphorylation [76]. Mammalian AMPK is activated through low concentrations of glucose due to rising AMP/ATP and ADP/ATP ratios [85]. The status of AMPK activation relies on the degree of energy deficiency [79]. In addition, AMPK may be activated by hypoxia, reactive oxygen species (ROS), and DNA damage [86,87,88]. In cattle, AMPK is closely associated with energy imbalance and heat stress. An energy imbalance in mammary tissue may lead to the accumulation of ADP or AMP, thereby activating AMPK. Heat stress may increase ROS levels and DNA damage, activating AMPK [89,90]. Previous studies have demonstrated the correlation between AMPK and milk fat. Specifically, Zhang et al. reported that 5-aminoimidazole-4-carboxamide ribonucleotide (AICAR), an activator of AMPK, suppresses the enzymatic action of ACC1 in goat mammary epithelial cells (GMECs) [91]. AMPK activation reduces the activity of lipogenic genes, including FAS and FABP3, in BMECs [92], which indicates that AMPK suppresses the de novo biosynthesis of FAs. Fan J et al. [93] identified PPARG, hormone-sensitive lipase (HSL), and SREBP1 as downstream genes in the AMPKα1 signaling pathway during lipogenesis. AMPK has been demonstrated to inhibit the transactivation efficacy of SREBP1c via the phosphorylation of SER372, thereby inhibiting the biosynthesis of milk fat [94]. Accordingly, AMPK indirectly controls the transcription of lipogenic genes in the mammary gland via SREBP1c phosphorylation. Shi H et al. [95] reported that using dorsomorphin, an inhibitor of AMPK, increased the transcription expression of ChREBP and FASN, indicating that glucose-mediated AMPK negatively regulates the de novo biosynthesis of FAs. These findings confirmed that glucose-mediated de novo FA biosynthesis in goat mammary glands relies on the AMPK-ChREBP axis, with ChREBP serving as one of the downstream pathways of AMPK. The structure, functional description, and interaction with conjugated linoleic acid of AMPK are shown in Figure 3.

### 2.4. Peroxisome Proliferator-Activated Receptor Gamma (PPARG)

PPARG is a member of the nuclear hormone receptor superfamily [96] and performs vital functions in adipocyte differentiation, carbohydrate metabolism, and lipid metabolism [12]. It forms heterodimers with retinoid X receptors (RXRs) and binds to PPAR response elements (PPRE sites) after ligand activation, thereby regulating transcription [97,98]. Shi et al. [99] knocked out the PPARG gene in GMECs. They reported a decrease in the expression of genes related to FAs biosynthesis and triglyceride biosynthesis, namely diacylglycerol acyltransferase 1 (DGAT1), SREBF1, ACACA, FASN, FABP3, and perilipin 3 (PLIN3). These findings indicate that PPARG is crucial for modulating milk fat biosynthesis and secretion. It has been reported that the relative mRNA levels of PPARG1 and PPARG2 in buffalo mammary glands are greater during lactation than during the dry period, indicating that PPARG1 and PPARG2 perform vital functions in milk fat biosynthesis during lactation [100]. Yu et al. [101] demonstrated that the PPARG gene regulates lipid accumulation in goat mammary cells by modulating the expression of the adipocyte differentiation-related protein PLIN2. Kast-Woelbern et al. [102] highlighted the relationship between PPARG and SREBP in murine adipocytes, according to which PPARG indirectly affects the activity of the SREBP protein by modulating the expression of INSIGs. Zhou et al. [103] reported that PPARG inhibition and overexpression in buffalo had different effects on the expression of genes, namely CD36, FABP3, fatty acid binding protein 4 (FABP4) (FA transporter), ACSS2, elongation of very long-chain fatty acid protein 6 (ELOVL6) (FA activation and elongation), LPIN1, diacylglycerol acyltransferase-2 (DGAT2), AGPAT6, butyrophilin subfamily 1 member A1 (BTN1A1) (triglyceride and lipid droplet formation), PPARGC1A, INSIG1, INSIG2, INSIG2, FASN, and SREBF2 (regulation of milk fat synthesis). In primary bovine mammary alveolar cells, treatment with a PPARG agonist promoted the expression of AGPAT6 and DGAT1 [3]. PPARG binding sites are likewise present in the promoter region of ELOVL6 in goats [104] and the transcriptional regulatory region of fatty acid desaturase 2 (FADS2) in humans [105]. In summary, SREBF2, AGPAT6, DGAT1, ELOVL6, and other genes are regulated by PPARG. Previous studies have demonstrated that PPARG binds directly to PPREs to boost the performance of the PLIN2 promoter [101]. Tian et al. reported that overexpressing the CCAAT enhancer-binding protein α (C/EBPα) promoted the mRNA expression of PPARG, indicating that PPARG is a downstream target gene regulated by goat C/EBPα [106]. Accordingly, C/EBPα modulates the expression of genes involved in lipid metabolism through the C/EBPα-PPARG pathway. These two transcription factors (C/EBPα and PPARG) exert synergistic effects on the regulation of the expression of lipid metabolism genes [107]. Fan et al. [108] performed Kyoto Encyclopedia of Genes and Genomes (KEGG) pathway enrichment and reported that the PPAR signaling pathway included nine DEGs (Acyl-CoA synthetase long-chain family member 4, phospholipid transfer protein, adiponectin, cytochrome P450 family 27 subfamily A member 1, perilipin 1, solute carrier family 27 member 4, aquaporin 7, Acyl-CoA oxidase 3, and perilipin 4 (ACSL4, PLTP, ADIPOQ, CYP27A1, PLIN1, SLC27A4, AQP7, ACOX3, and PLIN4), exhibiting significant differences between the middle lactation and late lactation stages (*p* < 0.05). The structure, functional description, and interaction with conjugated linoleic acid of PPARG are shown in Figure 4.

### 2.5. Phosphatidate Phosphatase 1

Phosphatidate phosphatase 1 (LPIN1) belongs to the lipin protein family. In mammals, LPIN comprises no less than one nuclear localization sequence (NLS), one N-terminal N-LIP domain, and one C-terminal C-LIP domain2. The three subtypes of LPIN are LPIN1 and phosphatidate phosphatase 2 and 3 (LPIN2 and LPIN3), which encode the lipin1, lipin2, and lipin3 proteins, respectively [109,110]. Lpin1 and Lpin2 exhibit differential expression in different tissues. While Lpin1 exhibits the highest expression in skeletal muscle and adipose tissue, the highest expression of Lpin2 is detected in hepatic tissue and brain tissue [111]. Han et al. confirmed that in yeast and humans, Lpin1 exhibits a PAP activity [112]. In addition, mouse Lpin1, Lpin2, and Lpin3 have been identified as PAP enzymes, with the PAP activity of Lpin1 markedly exceeding that of Lpin2 and Lpin3 [113]. LPIN1 catalyzes the dephosphorylation of PA to diacylglycerol (DGA) during triglyceride synthesis, thereby promoting the synthesis of TGs and phospholipids. In addition, Mg^2+^-dependent lipin serves as a transcriptional coactivator in the modulation of the expression of genes associated with lipid metabolism [114]. Finck et al. demonstrated that in hepatocytes, Lpin1 promotes the expression of the target genes of PPAR-gamma coactivator 1α (PGC-1α)/PPARα, which are engaged in FAO. This process includes acyl-CoA oxidase, PPARα, carnitine palmitoyltransferase-1, and medium/long-chain acyl-CoA dehydrogenase and inhibits the expression of genes associated with de novo lipogenesis, namely SREBP1, FASN, and SCD. Lpin1 has also been reported to cooperate with hepatocyte nuclear factor 4 α (HNF4α), PPARβ, PPARG, and the glucocorticoid receptor [115]. Moyes et al. [116] confirmed that LPIN1 is a downstream gene in the PPARG signaling pathway in bovine mammary glands. Kim J et al. [117] reported that overexpressing LPIN1 in 3T3-L1 cells increased PPARG expression and lipid droplet formation while silencing the LPIN1 gene reduced PPARG agonist-induced adipogenesis and diminished the phosphorylation levels of both PPARG and extracellular signal-related kinases 1 and 2 (ERK1/2). A lack of LPIN1 could inhibit normal triglyceride accumulation and induce the expression of lipogenic genes, namely PPAR and C/EBP [118]. The above studies demonstrated that the lipin family serves a bidirectional governing role in fat metabolism. Furthermore, studies have demonstrated that the LPIN1 gene mRNA is upregulated 20-fold in lactating dairy cows [119] and 1.39-fold in lactating buffaloes [120]. LPIN1 mRNA is positively correlated with milk yield in buffaloes and negatively correlated with milk fat content [121]. The gene expression of LPIN1 is governed by cholesterol. SREBF1 binds directly to the sterol regulatory element (SRE) in the LPIN1 promoter region, thereby promoting LPIN1 transcription [122]. LPIN1 is additionally modulated by SREBF1, estrogen-related receptor γ (ERRγ), HNF4a, C/EBPα, and other transcription factors in both human and mouse models. These transcription factors bind directly to their respective sites in the LPIN1 promoter region, enhancing LPIN1 transcription and lipid biosynthesis [36,123,124,125]. Certain studies have identified loci associated with milk production traits in the LPIN1 gene in dairy cattle [126] and yaks [127]. Zhou F et al. reported that simultaneous mutations at the PPRE1 and PPRE2 sites markedly decreased the function of the core promoter compared to that noted when PPRE1 or PPRE2 was mutated alone. The expression level of LPIN1 decreased after PPRE1 and PPRE2 mutagenesis [128]. These findings indicate that PPARG instantly regulates the transcription of buffalo LPIN1 by interacting with the PPRE1 and PPRE2 sites in the LPIN1 promoter region. The structure, functional description, and interaction with conjugated linoleic acid of LPIN1 are shown in Figure 5.

### 2.6. Thyroid Hormone-Inducible Hepatic Protein (THRSP)

The thyroid hormone-inducible hepatic protein (THRSP) gene encodes the Spot14 (S14) protein, which is associated with regulating the de novo biosynthesis of FAs in hepatic tissue, adipose tissue, and lactating mammary glands [129]. The main location of the THRSP protein is the cell nucleus. The THRSP protein can be rapidly regulated by metabolic fuels and fuel-related hormones and is associated with FA synthesis in rodents [130]. The THRSP gene of cattle is found on chromosome 29 (Bta29) and is 1398 bp in length, containing two exons. Polymorphism of the THRSP gene has been studied in all kinds of livestock. In chickens, the THRSP gene is repetitive and has two forms: thyroid hormone-inducible hepatic protein α and β (THRSPα and THRSPβ). In Qinchuan cattle, various THRSP genetic types are associated with water retention and tenderness, while these genotypes are associated with unsaturated, monounsaturated, and minor singular FAs in Hanwoo beef [131]. In Italian H-F dairy cattle, different THRSP alleles are correlated with milk yield traits [132]. Polasik et al. performed experiments on Jersey and Polish Holstein cattle and reported that the THRSP polymorphism was related to the proportion of FAs in milk fat. In addition, THRSP polymorphism significantly affects palmitic acid, stearic acid–hexanoic acid, lauric acid, myristic acid, palmitoleic acid, and FAs, which include 14, 16, and 6, respectively, through 16 carbon atoms [133]. Rudolph et al. [134] reported that overexpressing THRSP in the mammary epithelial cells of THRSP transgenic mice resulted in increased levels of C14:0. Cui et al. [135] reported that overexpressing THSP in BMECs augmented triglyceride levels and promoted the expression of lipogenic genes, such as FASN, PPARG, and SREBP1. Yao DW et al. [136] reported that overexpressing THRS in GMECs upregulated the expression of the relevant lipogenic genes, namely FASN, SCD1, DGAT2, and GPAM. However, the expression of the CD36 gene was downregulated, while the expression of ACACA and SREBF1 remained unaltered. Yao DW et al. [136] also reported that overexpressing THRSP led to a marked increase in the concentration of TGs and FAs with 12 and 14 carbon atoms, respectively. However, Salcedo-Tacuma et al. [137] experimentally demonstrated that in periparturient Holstein dairy cattle, THRSP acted as a suppressor of lipid biosynthesis in adipose tissue. In summary, THRSP is a regulator of dairy cow milk fat and gives rise to the increased synthesis of medium-chain fatty acids. However, whether THRSP stimulates milk fat biosynthesis in dairy cows is debatable among scholars, probably due to differences in species or gene transcription without protein expression. The structure, functional description, and interaction with conjugated linoleic acid of THRSP are shown in Figure 6.

The common pathways associated with the modulation of milk fat biosynthesis are the mTOR, SREBP, PPAR, AMPK, Lpin1, and THRSP pathways. However, these pathways do not act individually; rather, they act in synergy with their upstream and downstream pathways. A few of the recently discovered milk fat synthesis pathways are outlined below.

### 2.7. Elongation of Very Long-Chain Fatty Acid Protein (ELOVL)

There are seven isoforms of elongation of very long-chain fatty acid protein (ELOVL) known as endogenous fatty acids: elongation of very long-chain fatty acid protein 1 (ELOVL1) to elongation of very long-chain fatty acid protein 7 (ELOVL7). Among these, ELOVL6 is responsible for elongating saturated fatty acids (SFAs) and monounsaturated fatty acids (MUFAs). ELOVL6 primarily contains FAs with more than 16 carbons, while fatty acids with fewer than 16 carbons are synthesized mainly through FASN. Consequently, ELOVL6 serves as a pivotal gene for sustaining LCFAs [138]. Chen et al. [139] studied the structure of ELOVL6 and validated three single nucleotide polymorphisms (SNPs) in ELOVL6. These three SNPs exhibited significant associations with milk yield in cattle, and the single nucleotide polymorphism 3 (SNP3) mutation was linked to fat content.

In mice, ELOVL6 exhibits high expression in various tissues that are critical for lipid metabolism, such as white adipose tissue, brown adipose tissue, skin, testis, brain, liver, and adrenal glands [140,141]. Furthermore, the increase in the transcription of this gene within the mammary gland tissues of goats and cattle underscores its function in milk fat metabolism. Fan et al. discovered that overexpression of ELOVL6 dramatically increased the expression of genes such as INSIG1, INSIG2, SREBP, PPARG, FASN, GPAM, DGAT2, and APGAT6. At the same time, ELOVL6 knockout dramatically decreased the mRNA abundance of the INSIG2, SREBP, FASN, SCD, GPAM, APGAT6, and tail-interacting protein 47 (TIP47) genes. This, in turn, resulted in corresponding changes in the total triglyceride content in the BMECs [142]. These findings indicate that ELOVL6 not only catalyzes the biosynthesis of LCFAs in BMECs but also indirectly regulates the expression of genes associated with milk fat biosynthesis to increase milk fat production in cattle. With the identification of genes associated with milk fat synthesis in cattle, new candidate genes for dairy cow milk fat synthesis include phosphatidylinositol 4-kinase type 2 alpha (PI4K2A), solute carrier family 16 member 1 (SLC16A1), ATPase phospholipid transporting 8A2 (ATP8A2), vascular endothelial growth factor D (VEGFD), inhibitor of DNA binding 1 (ID1) [143], and arachidonate 12-lipoxygenase (ALXO12) [144].

## 3. Regulation of Milk Fat Synthesis via CLA

### 3.1. CLA Inhibits Milk Fat

Oliveira RC et al. [145] administered 10 g/d CLA isomers (t10, c12-CLA and c9, t11-CLA) and an equal amount of inert rumen fatty acids as controls to prepartum dairy cows and observed that the cows supplemented with CLA exhibited lower serum FA levels on day 1 and day 7 after delivery, while the overall postpartum serum β-hydroxybutyrate (BHB) level was observed to be low. The LFAs needed for milk fat biosynthesis in dairy cows are derived from plasma. BHB is also capable of directly condensing with acetyl-CoA, generating 3-hydroxybutyryl CoA. 3-Hydroxybutyryl CoA is subsequently reduced to butyryl CoA, which acts as a substrate for fatty acid biosynthesis. Therefore, decreasing postpartum serum FA and BHB levels could decrease milk fat intake and relieve NEB by reducing energy output. This inference indicates that prepartum CLA supplementation benefits postpartum performance. Rahbar B et al. [146] administered 120 g/d of rumen-protected CLA as a supplement to lactating Holstein cows and reported that CLA increased lactose and diminished milk fat and the concentrations of SFAs and MFAs. A blood sample analysis revealed increased amounts of glucose, cholesterol, TGs, insulin, insulin-like growth factor-1 (IGF-1), estradiol, and progesterone in the serum, while the serum beta-hydroxybutyric acid (BHBA) levels decreased. Decreased TG and BHBA lead to decreased milk fat biosynthesis, reduced energy output, and increased serum glucose concentration. Granados-Rivera et al. [147] administered different doses of conjugated linoleic acid to 15 early lactating goats and reported that the milk fat yield and milk fat composition decreased while the milk yield increased. This could have occurred because the surplus energy used for milk fat production was utilized to increase the milk yield. Dirandeh et al. [148] also reported that supplementing dairy cows with CLA could reduce the fat content and yield of cow’s milk while promoting the body condition of the cows. CLA reduces milk fat generation, decreases fat mobilization, and relieves the negative energy balance. Qin et al. [149] fed a mixture of CLA to early lactating dairy cows and reported that the milk fat composition and yield were significantly decreased in early and mid-lactating dairy cows. Furthermore, CLA was observed to confer insulin resistance. In summary, a certain dose of CLA may inhibit milk fat synthesis. The reason behind the different dosages of t10, c12-CLA is that the purities of rumen-protected t10, c12-CLA products manufactured at different coating companies are different, which leads to inconsistent dosages. Table 1 summarizes the mechanisms by which different doses of CLA affect milk fat synthesis.

### 3.2. The Mechanism through Which CLA Inhibits Milk Fat 

Several studies have demonstrated that CLA may diminish the milk fat content by providing calcium salts [150,151] or through direct rumen infusion [152,153]. Certain trials have identified the underlying molecular mechanisms, which include the reduced mRNA expression of fatty acid synthases [154]. These fatty acid synthases mainly include LPL, ACACA, FASN, and SCD. In studies where t10, c12-CLA was infused into the bovine rumen, the mRNA expression levels of ACC, FAS, and SCD1 in the mammary gland decreased [155,156]. Researchers infused CLA into dairy cows, followed by northern blotting to study gene activation in the mammary tissue of dairy cows. After 5 days of CLA infusion, the mRNA expression levels of the fatty acid synthase genes FASN, ACACA, and SCD were downregulated by 40%, 39%, and 48%, respectively. The mRNA expression levels of the LPL and FABP genes decreased by approximately 50%. In addition, the mRNA expression levels of the glycerol phosphate acyltransferase (GPAT) and AGPAT genes decreased by 42% and 41%, respectively [157]. After treating MAC-T cells and primary BMECs with t10, c12-CLA, ACACA, FASN, and SCD1, the mRNA and protein expression levels markedly decreased. In in vivo infusion experiments and in vitro models for t10, c12-CLA exhibited reduced mRNA expression of ACACA, FASN, and SCD1 in primary BMECs and MAC-T cells. After silencing the SREBP gene with siRNA, the expression of ACACA, FASN, and SCD1 was observed to be strongly downregulated [158]. Kadegowda et al. treated Mac-T cells with various long-chain FAs and reported that t10, c12-CLA and t10 18:1 inhibited the gene expression of FASN, SCD, and SREBP1, and t10, c12-CLA inhibited the gene expression of ACSS2, FABP3, INSIG1, SREBP2, and THRSP [3]. Similar studies have also demonstrated that t10, c12-CLA regulates the gene expression and activity of several lipometabolic enzymes, including SCD, CPT, lipoprotein lipase-1 (LPL1), and malic enzyme (ME). These enzymes participate in the uptake, transport, de novo biosynthesis, and unsaturation of FAs and the biosynthesis of TGs. According to previous studies, t10, c12-CLA is more effective than c9, t11-CLA at increasing CPT activity in male rat adipose tissue and promoting FAO. CPT is the rate-limiting enzyme in fatty acid β-oxidation that promotes the breakdown of TG within cells and reduces FA production [159]. Zhang et al. stimulated GMECs with t10, c12-CLA and observed that the mRNA abundance of FASN, SCD1, and ACSL4 decreased as the t10, c12-CLA concentration increased, as revealed through PCR and Western blotting analyses. The protein levels of FASN and SCD1 and the phosphorylation of AMPK and ACACA also decreased [160]. Zhang et al. previously reported that AMPK activation increases CPT1 mRNA levels and decreases FASN and ACACA mRNA levels [91]. In addition, AMPK has been confirmed to regulate energy metabolism in MECs. The above research indicates that t10, c12-CLA inhibits the activity of ACACA, FASN, SCD1, and ACSL4 in GMECs through AMPK activation.

The evidence suggests that a relationship exists between t10, c12-CLA and SREBP, according to which t10, c12-CLA modulates SCD1 by impacting the binding of the SREBP1 protein to the SRE and NF-Y sites in the SCD1 promoter region, thereby inhibiting fatty acid desaturation. In addition, the expression of lipogenic genes, namely FASN, ACACA, and SCD1, was reported to decrease [161]. Chen et al. [162] reported that t10, c12-CLA inhibits the interaction between ubiquitin-like protein 8 and INSIG1 and reduces INSIG1-mediated proteasome degradation, ultimately inhibiting the activation of SREBP1 in BMECs. Sandri et al. [163] administered PPARG agonists and CLA to lactating ewes and reported that CLA decreased the expression of SREBP1, SCD1, and mTOR in the mammary gland, and PPARG agonists could not reverse the effect of CLA, indicating that CLA inhibits milk fat biosynthesis via a signaling pathway independent of PPARG. Different studies have reported different dosages of t10, c12-CLA, possibly for different reasons both in vitro and in vivo. In vivo, the reason is that the purity of rumen-protected t10, c12-CLA products manufactured at different coating companies is different, which leads to inconsistent dosages. In vitro, t10, c12-CLA belongs to a class of lipids that do not dissolve directly in culture media when used for cell stimulation, and this issue can be resolved by adding different solvents during the experiments, which results in different drug concentrations. The interaction between conjugated linoleic acid and genes, proteins, pathways, and transcription factors related to milk fat synthesis is shown in Figure 7.

### 3.3. Modulation of Milk Fat Synthesis via Functional Supplements 

The suppressive effect of CLA on milk fat is widely recognized. Other substances may also exert effects, either similar or opposite to the effects exerted via CLA. For instance, certain reports have suggested that melatonin acts as a fat regulator in mammals. Melatonin inhibits milk fat synthesis by decreasing the phosphorylation of mTOR, 4E-BP1, and p70S6K [164]. Wang et al. [165] stimulated bovine mammary epithelial cells with 0.5 mM nicotinic acid and the G protein-coupled receptor 109A (GPR109A) agonist and reported that nicotinic acid significantly increased the phosphorylation of AMPK and ACC while knocking out GPR109A eliminated the effects of nicotinic acid on AMPK and the ACC. These findings indicate that nicotinic acid inhibits fat synthesis in BMECs through downstream signaling pathways via the mediating effects of GPR109A. Li et al. [166] stimulated BMECs with a spectrum of concentrations of sodium acetate (SA). They reported that SA could be transported to BMECs through fatty acid transport protein 4 (FATP4), and cyclin-dependent kinase 1 (CDK1) expression increased. The PI3K-mTOR-4EBP1/S6K and mTOR-SREBP-1 signaling pathways were activated, ultimately resulting in the biosynthesis of milk fat. Additional studies have suggested that acetate esters promote milk fat biosynthesis by enhancing fatty acid biosynthesis in a dose-dependent manner [167]. Recently, studies have demonstrated that acetate esters promote the expression of ACCA and FAS proteins via the mTOR-S6K1 signaling pathway to increase milk fat biosynthesis [168].

The common plant extract β-sitosterol has been demonstrated to activate the Janus kinase 2/signal transducer and activator of transcription 5 (JAK2/STAT5) and mTOR signaling pathways when used at concentrations ranging from 0.1 to 10 µM in BMECs, thereby affecting the protein expression of SREBP1 and PPARG and upregulating the synthesis of β-casein and milk fat in BMECs [169]. Butyric acid, an SFA, is synthesized through the fermentation of carbohydrates in the rumen of ruminants [170]. Cheng et al. [171] reported that sodium butyrate could increase the nuclear translocation of SREBP1 through the G protein-coupled receptor 41 (GPR41)/AMPK/mTOR/S6K signaling pathway while increasing the acetylation of mature SREBP1 through the GPR41/AMPK/Sirtuin 1 (SIRT1) pathway to enhance milk fat biosynthesis. Methionine (Met) reportedly exerts a crucial modulatory effect on milk production and promotes milk fat biosynthesis. Qi et al. [172] stimulated BMECs with different concentrations of Met and reported that 0.6 mM Met significantly increased the expression of sodium-coupled neutral amino acid transporter 2 (SNAT2) in the cytoplasm of BMECs. These findings further confirmed that SNAT2 is an activator of PI3K and its downstream effector, mTOR, SREBP1c, and Cyclin D1 signaling pathways. Therefore, Met may promote fat biosynthesis in BMECs via the SNAT2-PI3K signal. Recently, Lin et al. [173] studied bovine mammary epithelial cells and reported that Met activates PI3K, thereby reducing the mRNA and protein expression of AT-rich interaction domain 1B (ARID1B). Met may induce ARID1B degradation through the ubiquitin–proteasome system, thereby reducing the binding of ARID1B to the mTOR promoter. These findings and other reports suggest that amino acids such as Met could stimulate mTOR to promote milk fat biosynthesis. In addition, according to these studies, ARID1B could inhibit Met-induced mTOR activation to suppress fat and protein synthesis in MECs. Additionally, Ding L [174] reported that arginine, when administered through intravenous injection to cows, increased the concentration of NO in the blood, which led to increased blood flow and nutritional supplementation in the mammary gland, thereby promoting the resynthesis of FAs and milk production in the mammary gland. The study also revealed that the main pathway affected in this study was the AMPK signaling pathway. Moreover, Li et al. [175] reported that CREB-regulated transcription coactivator 2 (CRTC2) acts as a crucial facilitator of amino acid-induced milk fat biosynthesis in MECs. The specific mechanism reported by these authors involved the facilitation of bovine milk fat biosynthesis using amino acids through the mTOR-CRTC2-SREBP1c signaling pathway. *Camellia oleifera* Abel. Seed oil is a commonly used edible oil in China and is reported to exert a crucial modulatory effect on milk fat and milk protein biosynthesis in dairy cows. Zhong et al. [176] reported stimulating BMECs with *Camellia oleifera* Abel. Seed oil enhanced the expression of the SREBP1 gene and induced the expression of the FASN, ACC, and LPL genes, indicating that *Camellia oleifera* Abel. Seed oil could promote the de novo biosynthesis of FAs. The specific mechanism involved implies that the increase in mRNA and protein (including phosphorylated) levels within the PI3K-AKT-mTOR and JAK2-STAT5 signaling pathways leads to the regulation of related lipogenic genes, which ultimately leads to milk fat synthesis. Table 2 summarizes the effects of different functional supplements on milk fat.

## 4. Conclusions

In summary, the main mechanisms of milk fat synthesis include the de novo synthesis of short-chain fatty acids (FASN, ACACA, and SCD); the uptake, transport, and activation of long-chain fatty acids (LPL, VLDLR, ACSL, FABP3, and SCD); the synthesis of TGs (GPAM, LPIN1, DGAT, and AGPAT6); signaling pathways; and transcription factor (mTOR, SREBP, PPARG, and AMPK) production. The newly discovered genes involved in milk fat synthesis include ELOVL6, METTL3, THRSP, and candidate genes for milk fat synthesis: PI4K2A, SLC16A1, ATP8A2, VEGFD, ID1, and ALXO12. Currently, the research into the mechanism of milk fat synthesis mediated by CLA is limited to individual genes or signaling pathways. This article not only reviews the mechanism of action of individual genes or pathways in milk fat synthesis but also summarizes the possible relationships between each gene or pathway and the relationships between SREBP, PPARG, and LPIN1 (the binding site containing SREBP and PPARG). METTL3 regulates milk fat through mTOR and SREBP, and PPARG precisely regulates long-chain fatty acid transport genes (CD36, FABP3, and FABP4) and triglyceride synthesis genes (LPIN1, DGAT2, and AGPAT6) for lipid droplet synthesis (BTN1A1). In the future, the specific role of CLA in the milk fat synthesis network can be determined by overexpressing or knocking out genes or transcription factors and then comparing the results after using CLA. These findings can further provide a reference for the mechanism through which CLA inhibits milk fat synthesis. We summarize the mechanisms of action of substances other than CLA on milk fat, including genes or pathways such as melanin and nicotinic acid that reduce milk fat synthesis through mTOR, 4E-BP1, p70S6K, AMPK, and ACC; however, SA, β-sitosterol, sodium butyrate, Met arginine, and *Camellia oleifera* Abel increase milk fat biosynthesis through PI3K, mTOR-4EBP1/S6K, JAK2/STT5, mTOR, GPR41/AMPK/mTOR/S6K/SIRT1, SNAT2-PI3K, and other genes or pathways. The promotion or inhibition mechanism network of these substances on milk fat production can provide research references to further expand the mechanism through which CLA inhibits milk fat synthesis in cows.

## Figures and Tables

**Figure 1 animals-14-00204-f001:**
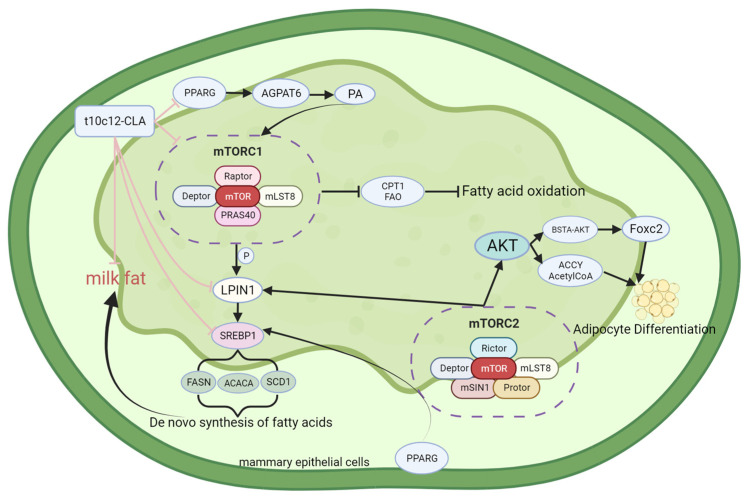
Structure and functional diagram of mTOR.

**Figure 2 animals-14-00204-f002:**
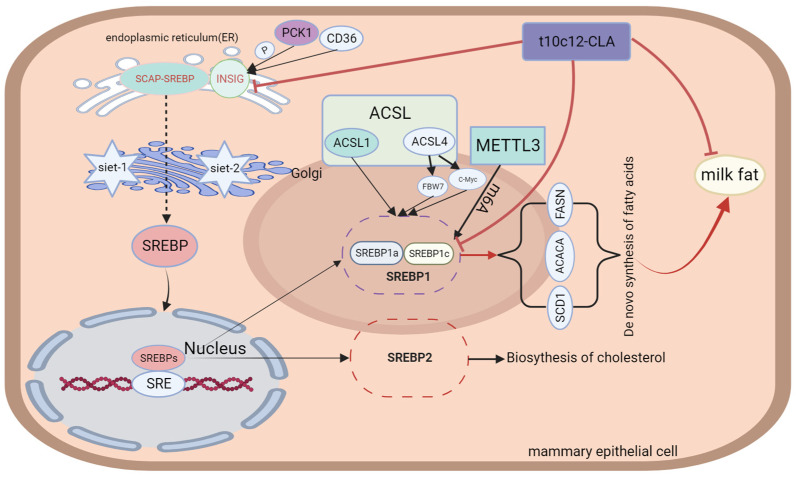
Structure and functional diagram of SREBPs.

**Figure 3 animals-14-00204-f003:**
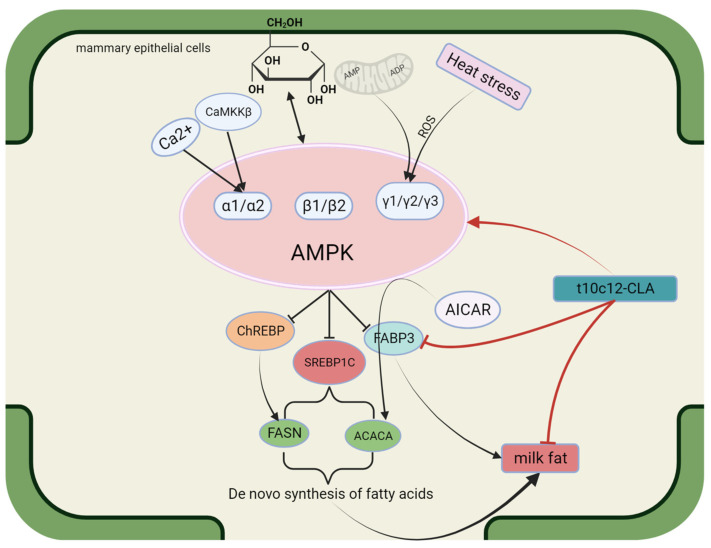
Structure and functional diagram of AMPK.

**Figure 4 animals-14-00204-f004:**
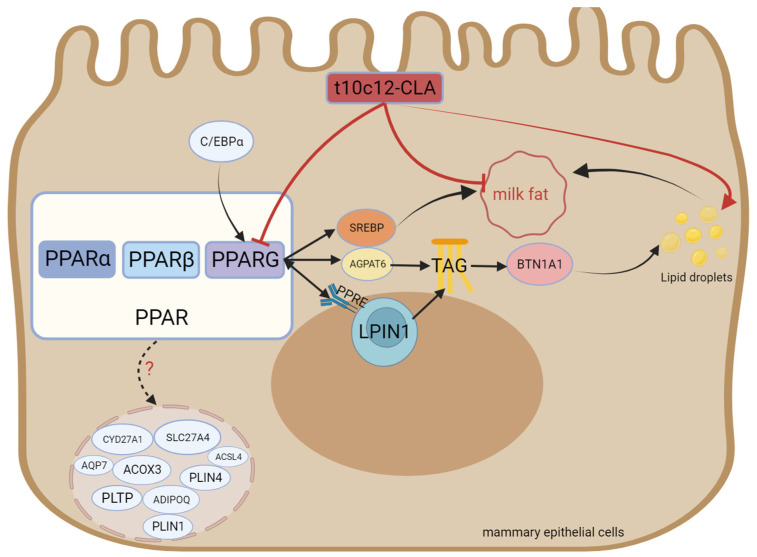
Structure and functional diagram of PPARG.

**Figure 5 animals-14-00204-f005:**
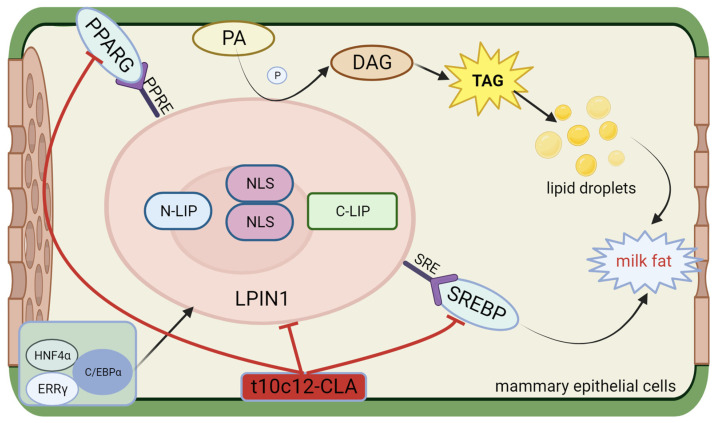
Structure and functional diagram of LPIN1.

**Figure 6 animals-14-00204-f006:**
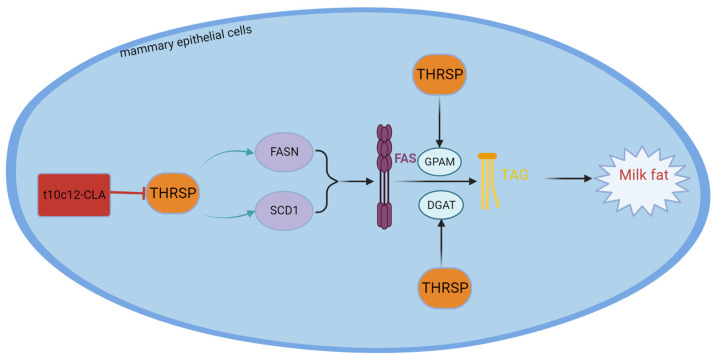
Structure and functional diagram of the THRSP.

**Figure 7 animals-14-00204-f007:**
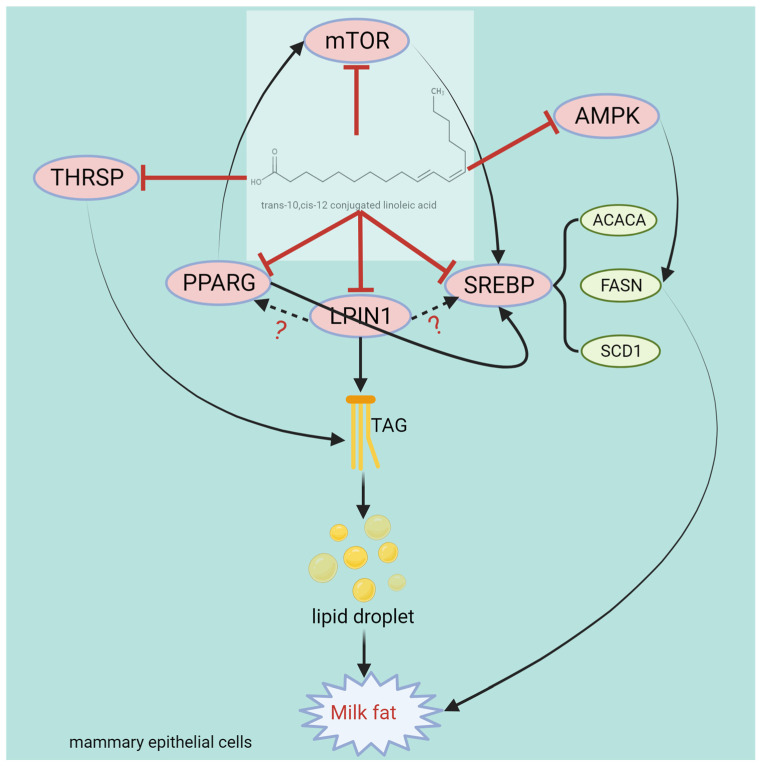
Structure and functional diagram of the t10, c12-CLA.

**Table 1 animals-14-00204-t001:** CLA inhibits milk fat.

Feeding Method	Dosage and Rumen Passing Form	Effect	References
Mixed with TMR (Total Mixed Ration)	10 g/d, CLA methyl ester	Elevated milk energy production; diminished postpartum serum FAs and serum β-hydroxybutyrate.	[145]
Mixed with TMR	50 g, 90 g/d, mixture of microencapsulated FA	Attenuated milk fat production and content; ameliorated energy balance.	[147]
Mixed with TMR	120 g/d, CLA of Lutrell^®^ Pure	Elevated lactose coupled with attenuated milk fat and concentrations of SFAs and MFAs; serum levels of glucose, cholesterol, TGs, IGF-1, estradiol, and progesterone increased, while serum β-hydroxybutyrate decreased.	[146]
Mixed with TMR	75/d, CLA of Lutrell^®^ Pure	Attenuated milk fat composition and yield, with potential improvements in metabolic health	[148]
Mixed with TMR	100 g/kg, CLA of Lutrell^®^ Pure	Milk fat composition and yield were markedly attenuated.	[149]

**Table 2 animals-14-00204-t002:** Regulation of milk fat synthesis via functional supplements.

Definition	Role	Result	References
melatonin	By attenuating the phosphorylation of mTOR, 4E-BP1, and p70S6K.	Reduced milk fat synthesis	[164]
nicotinic acid	Significantly increase the phosphorylation of AMPK and ACC.	Reduced fat synthesis	[165]
sodium acetate (SA)	Activating the PI3K mTOR 4EBP1/S6K and mTOR SREBP-1 signaling axis promotes the action of ACCA and FAS.	Increased milk fat synthesis	[166,167,168]
β-sitosterol	Activating the JAK2/STT5 and mTOR signaling pathways, thereby affecting SREBP1 and PPAR γ protein expression.	Increased synthesis of β-casein and milk fat	[169]
sodium butyrate	Increasing nuclear translocation of SREBP1 and acetylation of mature SREBP1 through the GPR41/AMPK/mTOR/S6K/SIRT1 signaling pathway.	Increased milk fat synthesis	[171]
Met	By stimulating the SNAT2-PI3K signaling pathway and reducing ARID1B binding to the mTOR promoter.	Increased milk fat biosynthesis	[172,173]
arginine	Increasing the concentration of NO in the blood and affecting the AMPK signaling pathway.	Promoted the resynthesis of FA and increased the production of milk	[174,175]
*Camellia oleifera* Abel.	Stimulated the PI3K-AKT-mTOR and JAK2-STAT5 signaling pathways, thereby promoting the action of SREBP1 and inducing the action of FASN, ACC, and LPL genes.	Enhanced the de novo biosynthesis of FAs	[176]

## Data Availability

Not applicable.

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
