# Peer review of "Research Progress on the Mechanism of Milk Fat Synthesis in Cows and the Effect of Conjugated Linoleic Acid on Milk Fat Metabolism and Its Underlying Mechanism: A Review"

_animals, 2024, doi:10.3390/ani14020204_

Round 1

Reviewer 1 Report (Previous Reviewer 2)

Comments and Suggestions for Authors

Through revision, the authors were able to address the concerns raised by the reviewer.

Comments on the Quality of English Language

Minor editing of English language required.

Author Response

1. Comments on the Quality of English Language: Minor editing of English language required.
AN: We thank the assigned editor for providing us with the contact information of the English editor. The manuscript was revised with the help of the English editor. (English Editing ID: english-75675)The revised manuscript has been submitted to the assistant editor (Jelena Babović) via email.

Reviewer 2 Report (Previous Reviewer 1)

Comments and Suggestions for Authors

1. Lines 515-517, please clarify what 'those differences‘ mean. Is it milk yield?

Author Response

1. Lines 515-517, please clarify what 'those differences‘ mean. Is it milk yield?

AN: Thank you for your valuable feedback. Here, we mainly want to explain the reasons why the usage of t10, c12-CLA varies in different studies. We have supplemented and clarified it clearly in the text.

“The reason behind the different dosages of t10, c12-CLA is that the purities of rumen-protected t10, c12-CLA products manufactured at different coating companies are different, which leads to inconsistent dosages”. (Lines 740-742)

This manuscript is a resubmission of an earlier submission. The following is a list of the peer review reports and author responses from that submission.

Round 1

Reviewer 1 Report

Comments and Suggestions for Authors

See the attachment for details.

Reviewer 2 Report

Comments and Suggestions for Authors

The review article ‘Effects of Conjugated linoleic acid (CLA) Supplementation on Milk Fat Metabolism and the Underlying Mechanisms in Dairy Cows: A Review’ reviews the literature for milk fat biosynthesis and the effect of CLA on some of the genes involved in biosynthesis. Overall, the review does not fulfill the objective stated in the abstract to ‘provide a brief summary of the existing comprehension of how CLA modulates milk fat biosynthesis in bovine mammary epithelial cells (BMECs) and the theoretical basis of the mitigation of NEB by CLA in perinatal dairy cows’ Suggestions for the manuscript are listed below.

Throughout the review, several abbreviations are not defined. As these are too numerous to list, only a few examples are provided. 

Lines 16 and 24, CLA is not defined in the simple summary or abstract where it is first used.

Line 74, a definition for DGAT is not provided until Line 304.

Lines 95-101, mTOR is defined in Line 95, but mTORC1 and mTORC2 are not defined.

Line 25 insert ‘and’ between ‘acid,’ and ‘accumulating’.

Line 35, the authors state that dairy cattle are important to China’s animal husbandry industry. Why are the authors limiting this statement to just China as dairy cattle are of economic importance in multiple countries.

Lines 43-45, the authors state that CLA’s are capable of reducing milk fat synthesis through downregulation of genes involved in breast fat synthesis. Why is this the only reference provided for CLA and fat synthesis? A search of Pubmed provides several publications that could have been referenced.

Line 52, what is the reason for using ‘etc’ twice in the sentence?

Lines 69-71, throughout the manuscript, the authors define abbreviations either before the abbreviation or after. Need to be consistent. 

Throughout the manuscript, the authors use PPARG (Line 126) and PPARg (Line 128). Assume these are the same. Need to be consistent.

Line 126, change ‘Intercede’ to ‘intercede’.

Table 1, what is TMR? This is never defined.

Lines 557 and 616, what is UnCLA? This is never defined.

The authors use 12 pages (page 2-13) to discuss anabolism of milk fat and only 4 pages (page 13-16) to discuss possible regulation of milk fat by CLA’s. Based on the title of the review and the objective of the review as stated in the abstract, the authors need to dedicate more time and space to the role of CLA’s in milk fat. For example, the authors need to include the role of CLA’s in the figures (Figure 1-7) and how CLA’s may regulate the genes in each figure. 

Comments on the Quality of English Language

Minor editing required. 

Reviewer 3 Report

Comments and Suggestions for Authors

Dear authors,

       the problem of CLA has been well documented. I have one question which is indirectly linked to your paper.

Dairy cows have been selected for decades for more milk and fat causing an increased negative energy balance. Now we are trying to reduced the NEB by feeding CLA. Is this option the correct way to deal with this problem?